# "If you find that I am HIV positive, don't tell me": Exploring the barriers and recommendations for HIV prevention services utilization among youth in rural southwestern Uganda

Paul Waswa Ssali[1], Timothy Mwanje Kintu[1], Immaculate Karungi[1], Agnes Kisakye Namuyaba[1], Tonny Kyagambiddwa[1], Ruth Namaseruka[1], Mark Agaba[1], Celestino Obua[1,2], Edith K. Wakida[2,3], Jerome Kahuma Kabakyenga[4]*

1 Faculty of Medicine, Mbarara University of Science and Technology, Mbarara, Uganda, 2 Office of Research Administration, Mbarara University of Science and Technology, Mbarara, Uganda, 3 Department of Medical Education, California University of Science and Medicine, San Bernardino, California, United States of America, 4 Department of Community Health, Mbarara University of Science and Technology, Mbarara, Uganda

* jkabakyenga@must.ac.ug

## Abstract

Globally, the majority of new HIV infections are recorded in Eastern and Southern Africa, with the youth being disproportionately affected. HIV prevention is the cornerstone of controlling the spread of HIV and ending this epidemic by 2030. However, barriers to the utilization of HIV prevention services remained underexplored especially among the youth in rural settings in sub-Saharan Africa. This qualitative study, conducted between February and April 2022 in rural southwestern Uganda, explored these barriers and identified recommendations to improve the utilization of HIV prevention services among youth. We conducted six focus group discussions (with youth [15–24 years] both in and out of school), nine in-depth interviews (with teachers, health workers, and members of the village health team), and four key informant interviews (with district officials) to collect data. Thematic analysis revealed barriers at the individual level (e.g., misconceptions, fear of testing, low perceived HIV risk, confidentiality concerns), community level (e.g., stigma, lack of counseling, peer influence), and health system level (e.g., lack of youth-friendly services). Recommendations included formation of youth peer support groups, ongoing awareness campaigns, and socio-economic empowerment initiatives, particularly targeting adolescent girls and young women. National scaling of these initiatives is essential to overcoming identified barriers and reducing HIV transmission among this vulnerable population. Additionally, economic empowerment especially among adolescent girls and young women in rural areas has enormous potential to address the spread of HIV in this sub-population.

**Data Availability Statement:** The datasets generated and/or analysed during the current study are publicly available as supporting information file.

**Funding:** This work was jointly supported by the Fogarty International Center (U.S. Department of State's Office of the U.S. Global AIDS Coordinator and Health Diplomacy [S/GAC] and the President's Emergency Plan for AIDS Relief [PEPFAR]) of the National Institutes of Health (under Award Number R25TW011210 to CO) and Mbarara University of Science and Technology (MicroResearch grant 3/ 2022 to JKK). The funders had no role in study design, data collection and analysis, decision to publish, or preparation of the manuscript.

**Competing interests:** The authors have declared that no competing interests exist.

# Introduction

Sub-Saharan Africa accounts for an estimated 69% of people living with HIV (PLWH) and an estimated 70% of AIDS-related deaths globally [1]. In 2021, the majority of the new HIV infections (44%) occurring worldwide were recorded in eastern and southern Africa [2]. Currently, Uganda ranks sixth globally among countries with the highest burden of new HIV infections, behind South Africa, Nigeria, Tanzania, Mozambique, and Kenya. In Uganda, young people, especially adolescent girls and young women, continue to be disproportionately affected by HIV, accounting for over one-third of new HIV infections [3]. To control the spread of HIV, a number of policies and HIV prevention programs have previously been designed, passed and implemented. These include ABC (Abstinence, Being Faithful, and Condom use), voluntary counseling and testing, access to anti-retroviral therapy (ART) for PLWH, and integration of HIV/AIDS issues in teaching, research, and service activities [4].

However, in a report by Uganda AIDS Commission (UAC), it was noted that there were insufficient efforts to address challenges with scaling of HIV prevention methods. The shortage of efforts to address structural issues (such as gender and economic inequalities and stigma) and behavior change interventions (such as community-based behavioral prevention and age-appropriate sex education) contribute to the burden of new HIV infections in the country [3]. Among other populations, a number of barriers to the utilization of HIV prevention methods have been identified. A previous study among adolescents and young adults living with HIV in Uganda reported that stigma, fear of HIV disclosure, fear of anti-retroviral therapy (ART) side effects, long waiting times, and low economic status were established as barriers to the utilization of HIV care services [5]. Among female sex workers in Uganda, stigma and economic status, breach of confidentiality, and stockout of drugs were described as barriers to the utilization of HIV prevention and care services [6].

The incidence of HIV is known to be higher in key populations such as the youth and fishing communities, which comprise the greater proportion of the district in which this study was done. Of note, the youth may be aware of the existence of some of these HIV prevention services, but myths and barriers may affect their utilization [7]. Additionally, it has previously been noted that HIV prevention services are mostly focused on urban hotspots in the country where new infections are expected, leaving out rural areas [3]. Therefore, this study explored the barriers to the utilization of HIV prevention methods among youth and established recommendations by stakeholders such as youth, health workers, and teachers to improve the utilization of HIV prevention methods in rural southwestern Uganda.

# Materials and methods

## Study design and setting

The study followed a descriptive approach, employed qualitative methods and was focused on exploring barriers and recommendations to the utilization of HIV prevention services among youth in rural South Western Uganda. This study defined barriers as any factor that hinders youth utilization of HIV preventive services.

The study site, Rubirizi district was chosen because it has the second highest prevalence of HIV among the youth in Uganda and the district has numerous fishing villages, which are a key HIV population. Particularly, a previous study indicated that HIV prevalence among the fishing communities in Uganda is estimated to be three times that of the general population [8]. It was therefore imperative that barriers to the utilization of HIV prevention services and possible solutions are extensively explored in such communities. The study included Focus Group Discussions (FGDs), in-depth interviews (IDIs) and key informant interviews (KIIs).

The study was broadly guided by the socio-ecological model (SEM). The SEM is a framework that examines the interplay between individual, community, and system factors in influencing health, emphasizing multi-level interventions for comprehensive health promotion. This framework has previously been utilized to describe barriers to utilization of HIV services [9].

## Study population

The study population included school and non-school-going youth aged 15 to 24 years regardless of their HIV status. The study also included teachers in-charge of health in schools, Village Health Team members (VHTs) or lay health workers and health workers directly involved in HIV care and treatment at the different health facilities. The key informants in this study were the district focal person for HIV/AIDS, District Health Officer (DHO), District Education Officer (DEO) and the District Community Development Officer (DCDO).

## Selection criteria

All youth in Rubirizi district aged 15–24 years, regardless of their educational status were eligible for the study. This encompassed both youth currently enrolled in schools within the district and those not attending school, at the time of the study. Health care workers at the HIV/AIDS clinic at the health centres in Rubirizi district and teachers in schools in Rubirizi district were also eligible. However, participants that did not give informed consent were excluded.

## Sample selection and data collection procedure

Following administrative approval from the DHO and head teachers, youth that were enrolled in school and teachers were recruited from conveniently selected government/public schools in Rubirizi district. Youth that were not enrolled in school at the time of the study were recruited using snowballing with the help of VHTs from purposively selected villages in the district.

In order to explore the barriers to utilization of HIV prevention services and suggest possible recommendations, FGDs were held with both youth in school and those who were not in school. Following previous recommendations, we planned for a minimum of five discussions and interviews [10]. FGDs were held until there was no new information emerging from the discussions. A total of six FGDs were held including 3 FGDs with youth in school and 3 FGDs with youth who were not in school. Each FGD in schools consisted of 8 participants that were randomly picked from all the classes with the help of teachers. Participants aged above 18 years were consented before the discussion whereas participants aged less than 18 years were assented following consent from the headteacher.

The FGD and interview guides were developed by the authors (PWS, TMK, TK and JKK), aligned with the study objectives and modified to fit the study setting. The FGDs were carried out in a quiet environment in school and church compounds. Questions in the FGD guide probed participants on their awareness of HIV preventive measures, and barriers to their utilization they had identified and explored possible recommendations. Each participant was given a unique identifying number to ensure confidentiality, and the discussions were audio recorded with one of the research assistants simultaneously taking notes. Youth that were not in school were selected using snowballing and FGDs were formed. Discussions were carried out by the research team (PWS, IK, AKN, RN, MA) at designated places like home to one of the participants or home to a VHT. FGDs lasted 45–60 minutes and participants were given refreshments and transport reimbursement at the end of the discussion.

There were three individual in-depth interviews were held with teachers, from each of the three schools, three with VHTs, and three with health workers that work in HIV clinics. These

interviews were carried out in a private environment after obtaining informed consent from the participants and lasted 45–60 minutes. The guide for the interviews was also developed by the research team. The interviews were mainly focused on the barriers to the utilization of HIV prevention services by the youth and suggestions of possible recommendations to overcome these barriers.

Four key informant interviews were held individually with the district health officer, the district community development officer, district education officer, and the district focal person for HIV. These informants were purposively selected because of their comprehensive understanding of the youth and their key positions in handling HIV services among the youth. The key informant interviews were done at their respective offices after informed consent was obtained and each lasted about 1 hour.

### Data management and analysis

All interviews were recorded and notes were taken by research assistants. Each recording was then transcribed verbatim in Runyankole, the local dialect before being translated into English by independent research assistants. The interviews/discussions conducted in English were transcribed verbatim. Both deductive (following the socio-ecological framework) and inductive (using thematic analysis as described by Braun and Clarke [11]) approaches were employed during the analysis. For thematic analysis, research team members familiarized themselves with the data through independent reading of each transcript. To ensure intercoder reliability, the researchers met to discuss and generate codes inductively from the transcripts. These codes were imported into Nvivo (QSR International) software to create a codebook. Subsequently, the codes were abstracted into deductively generated subthemes or categories (individual, community, and health system) and overarching themes (barriers to utilization of HIV preventive services and recommendations to increase utilization of these services).

### Ethics approval and consent to participate

Prior to data collection, the study protocol was reviewed by Mbarara University of Science and Technology research ethics committee (MUST-2021-291) for ethical clearance. Administrative clearance was obtained from the Chief Administrative Officer and the DHO. Following administrative clearance from the DHO, permission to carry out the study was obtained from the head teachers of the different schools for the school going youth. The study participants were then approached and study procedures explained before being asked to participate in the study. Prior to data collection, written informed consent was obtained from each participant older than 18 years; and assent was obtained from all study participants less than 18 years as well as informed consent from their guardians for those outside school and by the head teachers for those who were in school. Anonymity was ensured by not collecting participants' personal identifying information throughout the study.

### Results

Table 1 presents the profile of study participants stratified according to the data collection method used. There were six FGDs, nine in-depth interviews (IDIs) and four key informant interviews (KIIs) held.

Table 2 presents the participant characteristics for the FGDs and interviews. For the participants in FGDs who were all youths, there were equal numbers of males and females, the majority had attained at least primary level education and the majority of youth out of school had no

**Table 1. Profile of the study participants.**

| Data collection | NUMBER |
|---|---|
| **Key Informant Interviews participants** | **4** |
| *Position* | |
| District Health Officer | 1 |
| District Education Officer | 1 |
| District HIV Focal Person | 1 |
| District Community Development Officer | 1 |
| **In-depth Interviews** | **9** |
| *Position* | |
| Teachers in-charge of Health | 3 |
| Clinical Officer | 1 |
| ART treatment in-charge | 1 |
| Village Health Team members | 3 |
| Antenatal Nursing Officer | 1 |
| Focus Group Discussions | |
| **Youths** | **48** |
| Youth in School (YIS) | 24 |
| Youth out of School (YOS) | 24 |

**Table 2. Participant characteristics for focus group discussions & interviews.**

| | FGDs *(n = 48)* | KII & In-depth interviews *(n = 13)* |
|---|---|---|
| *Characteristic* | | |
| **Gender** | | |
| Male | 24 (50%) | 9 (69%) |
| Female | 24 (50%) | 4 (31%) |
| Youth in school | 24 (50%) | - |
| Youth out of school | 24 (50%) | - |
| **Marital status** | | |
| Married | 21 (44%) | 9 (69%) |
| Not married or divorced | 27 (56%) | 4 (31%) |
| **Highest education level** | | |
| None | 6 (13%) | 1 (8%) |
| Primary | 35 (73%) | 1 (8%) |
| Secondary | 5 (10%) | 1 (8%) |
| College/University | - | 6 (46%) |
| Postgraduate Education | - | 4 (31%) |
| **Religious affiliation** | | |
| Catholic | 31 (65%) | 7 (54%) |
| Anglican | 12 (25%) | 6 (46%) |
| Pentecostal | 1 (2%) | - |
| Seventh Day Adventist | 1 (2%) | - |
| Muslim | 3 (6%) | - |
| **Occupation** | | |
| Student | 24 (50%) | |
| Self employed | 4 (8%) | 1 (8%) |
| Subsistence farmer | 2 (4%) | 1 (8%) |
| None | 16 (33%) | 3 (23%) |
| Government Employee | 2 (4%) | 8 (62%) |

occupation while for the participants in KIIs and IDIs, the majority were males, married and government employees.

## Barriers to the utilization of HIV prevention services among youth

From the interviews and discussions with the youth, health workers, and teachers, a number of barriers to the utilization of HIV preventive services were explored. This study defined barriers as any factor that hinders youth utilization of HIV preventive services. The barriers were stratified into individual, community, and health system barriers.

## Individual-level barriers to the utilization of HIV prevention services

There were three individual-level barriers to the utilization of HIV prevention services identified. These included ignorance and misconceptions about HIV prevention services, fear of testing, low perceived susceptibility to HIV, and fear of breach of confidentiality.

## Ignorance and misconceptions towards HIV prevention services

It was noted during the discussions with the youth that a knowledge gap and false beliefs held about some methods of HIV prevention especially condom use played a big role in their under-utilization of these specific methods. One of the youths that was not in school had this to say:

> *"Even girls don't want boys to use condoms when having sex for the fear that they don't get sexual pleasure, that the boy may not put it on well and it enters deep in their vagina and cause vaginal cancer. All the above prejudices can't let them use condoms which results into easy spread of HIV/AIDS among the youth"* YOS FGD3

Surprisingly, even the youth in school also shared the same thoughts.

> *"So that's why I am saying that the use of condoms at some times its dangerous because you may find instantly or carelessly and you find that it has gone into the vagina then at times you are forced to go to the hospital to remove it or you find that you are forced to get cancer"* YIS FGD1

## Fear of testing for HIV among the youth

This was highlighted as an important barrier to HIV testing as a method of HIV prevention. The youth expressed fear of what comes after getting positive test results and therefore preferred to remain in the dark about their status.

> *"I told her and I advised her why can't you go for testing she refused telling me that if it were you, you can fear. After I forced her and went with her in the hospital, and we found that she had HIV. After she started.... she refused even take tablets and she ended up dying........."* YIS FGD2

> *"she told me that she is not feeling okay. Then after we had to say. Do you know what you're going to do you go at the hospital and test for HIV but besides she feared there[hospital]..."* YIS FGD3

> *"the youth asked me that 'if you find that I am HIV positive, what would you do to me and I said that I will declare the results to you and she said that if you find that I am HIV positive,*

*don't give me the result slip that is positive. Give the result slip that is HIV negative because my parents are going to ask me' . . ..so they are in that dilemma that when they are tested positive, it will be hard for them" IDI HW02*

### Low perceived susceptibility to HIV

It was evident during the discussions that some of the youth thought that they could not be affected by HIV which directly translated into under-utilization of the preventive services

*"The reason why some youth deep in the village don't test for HIV is that some of them are very proud of themselves. They just say that me to acquire HIV! No. . .they just have that feeling" YIS FGD1*

*"Majority of the youth take AIDS for granted and by the time they find that they have HIV, it rises stigma in them and fears to take tablets" YIS FGD2*

### Fear of breach of confidentiality by health workers

The youth reported that some health workers inform the rest of their community each time someone tests positive which stops them from accessing testing services or even going for counseling about HIV preventive services.

*". . .some youths don't go to test for HIV because workers there at the health Centers when you go to test and you find that your HIV positive, they spread rumors that this one is positive, hence affecting you" YIS FGD1*

### Community level barriers to utilization of HIV prevention services

At the community level, another three barriers to utilization of HIV prevention services emerged during the discussions with youth and teachers in charge of health in the schools. These were lack of counseling services in the communities, peer influence and stigma towards HIV.

### Lack of counseling services in the schools and community

Counseling services are an effective way of passing on information to the youth about HIV and guiding the youth on the use of HIV preventive measures. However, the youth cited that there was limited availability of counseling services both in the schools and the community.

*"For me, I can't lie, we don't have those laws in our school and counseling services" YIS FGD2*

Whereas other informants felt that counseling services were not readily available in the communities or schools they were available at health centers.

*"We normally do general counseling and also come to them directly for those youth living with HIV (YLHIV), so they end up picking clear messages and go to health centers for example Rugazi health center four and easily get medication from there with or without us unlike those ones in the villages who have nobody to counsel them, talk to them and many others" IDI T01*

*"We don't have any counseling services in our villages. Though if you find them at the nearby health centers, health workers can counsel you" YOS FGD3 number2*

Additionally, some teachers also felt that they were not well trained to handle HIV counseling

*"No, we don't do it, but basically, we do it on other diseases but not on HIV because of AIDS needs a trained counselor since us, we are used of punishing them, they may fear us to tell us most of the problems" IDI T02*

## Peer influence

It was reported that peer influence affects the utilization of specific HIV preventive services by the youth.

*"Then another reason why some youths don't go to test for HIV is because of peer group influence. You're my friend then I ask you why you go to test for HIV? there is nothing wrong with you and you can't go to test for HIV" YIS FGD1*

*"Some youth wait to see others going to test for HIV/AIDS so that they can also go there. Therefore, if their fellows don't go for HIV testing, they can't go there as well" YOS FGD3*

## Stigma towards HIV

The stigmatization that comes with testing positive from the community members, hinders utilization of most of these services.

*"Another thing is that stigma remains still a problem and because of this pandemic that have been there, it has put the youth on the high risk of acquiring HIV because there has been redundancy among youth and yet since they fear to be seen at the health facility . . ." IDI HW02*

## Health system barriers

Lack of youth-friendly services emerged as the only health system barrier to the utilization of these services.

## Lack of youth-friendly services

The HIV services at health facilities were described as not being favorable for the youth and favorable thereby hindering their utilization by this age category.

*"And another thing is that most of our health facilities we don't have youth friendly corners which means that we fail how to give youth services" IDI HW02*

*"Not at all because of two factors; one service and age go together because they also want their peer to serve them. So we are saying that it will be good if all health centers have a peer system especially office for the youth so that they know where they can meet their fellow youth and get services because it is still a general issue so there is still limited access of those youth out of school" KII 04*

## Recommendations for improving youth utilization of HIV prevention services

Study participants made several recommendations to overcome barriers preventing youth from utilizing HIV preventive services and these include; the formation of youth peer support groups, ongoing awareness campaigns, and socio-economic empowerment.

### Formation of youth peer support groups

Respondents highlighted the need to involve youth leaders in HIV prevention programs at the community level as they are likely to encourage youth involvement and utilization of HIV preventive methods. It was noted that these groups might make the mobilization of youth for HIV awareness campaigns easier for health workers to carry out.

"*I think there, they should use youth leaders in their community because they know themselves very well and prepare meetings and invite those medical workers to talk to them since most of the youth come from hard-to-reach areas that hinder them from getting services*". IDI T02

In addition, peer groups at the community level can also be employed at health facilities to work with health service providers in providing HIV services. The same groups of youth can be used at the community level to equip, encourage and provide guidance and counseling services to fellow youth as they seem to be more comfortable with their peers working on them. A health worker suggested,

"*We should also establish youth peer groups, and they work with us at the health facility and in the community because it is easy for a peer to encounter the youth at the village level.*" IDI HW02

Importantly, since youths were reported to be more comfortable being handled by fellow youth, training, and creation of youth teams that do work similar to that of the village health team members, but for their peers specifically was likely to get more youth involved in HIV campaigns. A key informant implied,

"*We should have community workers for the youth at the village level, not the VHTs. We should have mobile youth health workers for the youth, train them, and give them what is necessary like vehicles, Laptops, testing kits, CDS and move in the community to give services to the fellow youths in a humble way.*" KII 04

### Ongoing awareness campaigns

In line with the decentralization of health services to the lowest level in the community, some respondents reported that it was imperative to extend HIV prevention services to villages as some youths were less likely to seek such services that were only provided at parish levels and higher levels. Furthermore, it is also essential to include people from different campaigns while teaching about HIV preventive measures, as youth are less likely to listen to people they are used to seeing in the same locality. A member of the village health team stated,

"I *suggest that HIV education should be done at the village level because if it remains in the parish, the majority of the youth don't even go there, and we should normally have new faces*

*during that education because if they see us educating them, they don't take it seriously because they are used to us." IDI VHT02*

Furthermore, VHTs can be tasked to create relationships with the school administrations to give students more information about different methods of protection against HIV infection. In addition, this would make it easier for them to engage these youth out of school. This was proposed by a member of a village health team who stated,

*"For me, I think there should be a strong partnership between VHTs and schools so that we can educate the youth both at the school and out of school on how they should protect themselves from HIV/AIDS." IDI VHT03*

### Socio-economic empowerment

To address poverty-afflicted non-school-going youth in rural areas, it is essential to equip these young people with skills like carpentry, welding, and hairdressing. This could encourage them to start income-generating activities to take charge of their lives instead of seeking help from older adults that are likely to take advantage of them. A key informant suggested,

*"Economic empowerment if we can start some good programs that focus on youth like skilling. Yes, the youth could have left school at an early age, but they have hands, and they can go for saloons, carpentry, welding, and many others. So, they can be supported so they can get something to survive on other than depending on support from the non-related parents". KII 04*

Also, a health worker stated,

*"For those non-school going youth if they can be given income generating activities so that this system of going to the adult people yet they may be infected, so if they are given income generating activities, they may not go to seek for money from the older people." IDI HW02*

## Discussion

The main objective of this study was to explore barriers to the utilization of HIV prevention services among youth aged 15–24 years in rural southwestern Uganda and establish recommendations to improve the utilization of these services. These barriers were categorized into individual barriers (ignorance and misconceptions about the services, fear of testing, low perceived susceptibility to HIV, and fear of breach of confidentiality), community barriers (lack of counseling services in the communities, peer influence, and stigmatization), and a health system barrier (lack of youth-friendly services). Three main recommendations were established including the formation of youth peer support groups, ongoing awareness campaigns, and socio-economic empowerment.

It emerged that the youth held misconceptions and false beliefs about certain HIV prevention methods, especially the use of condoms, which negatively affected their utilization of these services. These misconceptions arose during discussions with both the youth in school and the youth out of school. Our findings are similar to those among adolescents in Nigeria [12] who also held a number of misconceptions are common about condom use. One explanation for these misconceptions is the absence of adult and parental guidance, leading youths to seek information from their peers, who may themselves be misinformed about HIV prevention

services. Furthermore, the rural setting of this study worsens the issue, as geographical isolation and limited internet connectivity restrict access to accurate and comprehensive health information. Additionally, the prevalence of traditional beliefs in rural areas regarding the causes and remedies for illnesses often contradicts scientific explanations of HIV prevention, further entrenching these misconceptions.

HIV testing services (HTS) have been described as an essential gateway to HIV prevention because early linkage to ART can prevent HIV transmission to one's partner and prevent vertical transmission [13]. Additionally, knowledge of one's status makes one more to choose an effective HIV prevention method. In this study, it was identified that the fear of testing positive affected the youth's utilization of HTS. This may be due to the associated stress that comes with testing positive as reported in a previous study conducted among refugee adolescents and youth in Kampala Uganda [14]. Additionally, the fear of testing could be due to the fear of being ostracized from the community (perceived stigma) which was also reported as a barrier to the utilization of these HIV prevention services in this population. Due to the fact that fear of testing is rooted in stigma [15], we suggest stigma reduction campaigns especially targeting the youth in rural communities. This is important because increased awareness has been reported to lead to changes in stigmatizing attitudes and an increase in HIV testing [16]. Additionally, we recommend widespread implementation of HIV self-testing (HIVST), especially in this age group. Previously, HIVST has been reported to increase the uptake and frequency of testing for youth and may tackle the challenge of fear of testing in this youth age group [17–19].

Low perceived risk of susceptibility to HIV was identified as a barrier to the utilization of certain HIV prevention services especially HIV testing and condom use. One's perception of their susceptibility to HIV infection is very important because it affects one's likelihood of getting involved in risky sexual behavior such as not using a condom [20]. Our findings are different from those of a quantitative study done in 2004 among adolescents in Uganda that found most female adolescents to have a high perceived risk of susceptibility to HIV [21]. However, this study used data from 2004, a period during which the mortality rate due to HIV was high, and HIV was therefore highly feared and the risk of susceptibility possibly higher. To address this low perceived risk, this youth population may therefore benefit from continuous awareness campaigns on the dangers of HIV.

This study also found that a shortage of counseling services in schools and communities affected the utilization of HIV prevention services. The study participants reported that these services are only at specific health facilities rather than at a more decentralized level (like villages) which isn't accessible to all. Additionally, teachers expressed the lack of skills in HIV counseling to guide students on the utilization of some of these HIV prevention methods. comprehensive sexuality education (CSE) has previously been suggested as a modality to deliver information on sexual and reproductive health to adolescents and youth in schools [22]. However, it is emerging that teachers in rural schools may not have the skills to deliver CSE. This is possibly due to the recent changes in CSE beyond abstinence-only education [23]. Therefore, we recommend the implementation of school-based CSE training for teachers, especially in rural areas that have been postulated to increase awareness of sexual and reproductive health [24].

We found that peer influence is one of the barriers to the utilization of HIV preventive measures. This is because the youth reported dependence on the opinions of their counterparts to guide their choice (or not) of usage of specific HIV preventive methods. This highlights the need to increase focus on peer education as an intervention to improve the uptake of HIV prevention services. The formation of peer groups typically involves recruiting members of a specific at-risk group to encourage members to change risky sexual behaviors and maintain healthy sexual behaviors [25]. Peer education is documented to enhance interpersonal

interaction allowing for more open discussions on sensitive topics such as sexuality [26]. Peer education also tackles the challenge of health workers not being in a position to reach them [24].

Health workers also reported that the lack of youth-friendly services at rural health facilities affected the youth's utilization of HIV prevention services. This lack of youth-friendly services has also been reported in a recent systematic review [27] on access and utilization of youth-friendly sexual and reproductive health services in sub-Saharan Africa. This review reported that the negative attitude and the lack of skills among health workers were some of the barriers to the implementation of youth-friendly services. Therefore, health workers may benefit from continuous training to equip them with the skills to provide youth-friendly services. Additionally, as has been suggested previously [28], youth in rural areas should be actively involved in the design and implementation of sexual and reproductive health services in order to ensure that they are youth-friendly. Finally, it has been suggested that HIV prevention services should be made as affordable as possible to ensure their utilization by the youth [27].

Three recommendations were reported by the stakeholders to address the noted barriers to the utilization of HIV prevention services. First and foremost, the need for ongoing HIV prevention awareness campaigns in rural communities. These awareness campaigns should be strategically targeted toward the youth and family structures at the village level and in the different schools. This is important because parents can be better informed to guide their children on the usage of different HIV prevention methods. These public awareness campaigns have been documented to influence behavior change and encourage openness, especially among the youth [29]. Intensification of the campaigns should be directed towards providing information on different HIV prevention methods, especially novel methods like pre and post-exposure prophylaxis, and also addressing previously identified common myths and misconceptions [30] about the different HIV prevention methods as identified by the study participants.

Another recommendation was the establishment of youth peer groups in rural communities. Peer support has been described to buffer the negative effects of stigma and positively influence behavior [31]. As aforementioned the youth are more likely to approach peers for HIV-related information as compared to adult healthcare workers. These peer groups could also play a role in ensuring sexual and reproductive health services at the community level are youth-friendly It has been noted that for sexual and reproductive health services to be youth-friendly, the youth should play a role in their design and implementation [24]. These groups could support the dissemination of information on HIV prevention methods to the youth of these HIV prevention services. However, the implementation and scale-up of these peer support services have been affected by insufficient funding and coordination in sub-Saharan Africa [32]. Therefore, this needs to be explored by the different stakeholders and policymakers.

Finally, economic empowerment was a recommendation given by health workers and district officials to help control the spread of HIV among the youth. Notably, adolescent girls and young women in high-prevalence regions, such as fishing villages in the Rubirizi district, are disproportionately affected by HIV [3]. These groups are particularly vulnerable to engaging in transactional and cross-generational sexual relationships, driven by limited economic opportunities [33, 34]. Previously described economic empowerment interventions include microfinance, vocational skills training, business development training, and micro-enterprise development [35, 36]. Economic empowerment has also been documented to have the potential to address other sexual and reproductive health challenges such as domestic violence by giving young women the economic independence to leave abusive relationships [37]. Our

findings highlight the need to actively scale up economic empowerment, especially in rural areas as a long-term mitigation strategy for the management of HIV spread.

One strength of this study is the purposive selection of participants, which included district administrative officials, teachers, healthcare workers, and youth (both in and out of school). This diverse mix allowed for a comprehensive exploration of barriers and recommendations for improving HIV prevention service utilization from various perspectives. In-depth interviews with healthcare workers and teachers provided a detailed understanding of these barriers. The engagement of different participant cadres and triangulation of data sources (FGDs, IDIs, KIIs) ensured the credibility and confirmability of the findings. All interviews were moderated by impartial research assistants that were not part of the community being studied, enabling open discussion. Uniform study guides were used to maintain the validity of collected information. Additionally, rich descriptions of the research context, including participant details and study settings, were provided to ensure the transferability of the findings.

The main limitation of this study is that barriers were explored from a small youth population, therefore there's a need to do quantitative studies to identify barriers to utilization with the largest magnitude to guide the prioritization of interventions by the different policymakers and stakeholders. Additionally, information on pre-exposure prophylaxis, an important biomedical HIV prevention method, was not explored in this study. As mentioned earlier, one of the recommendations, as solicited from the study population was to intensify campaigns emphasizing the availability of preventive methods like pre-exposure prophylaxis to safeguard themselves.

## Conclusion

The youth remain a key population in the struggle against HIV, and this study highlighted barriers to the utilization of HIV prevention services among youth in rural southwestern Uganda. These barriers were stratified at different levels including individual barriers (ignorance and misconceptions about the services, fear of testing, low perceived susceptibility to HIV, and fear of breach of confidentiality), community barriers (lack of counseling services in the communities, peer influence, and stigma towards HIV), and a health system barrier (lack of youth-friendly services). Additionally, three main recommendations were highlighted by the study participants that is, formation of youth peer support groups, ongoing awareness campaigns, and socio-economic empowerment. Some of the highlighted barriers (such as low perceived susceptibility to HIV) need to be further explored by researchers and other relevant stakeholders to develop a deeper understanding of their origins and possible solutions. Suggestions like nationwide scaling of youth peer support services have the potential to address specific barriers like stigma and improve uptake of specific HIV prevention methods. These findings highlight the need for concerted efforts from policymakers especially the government of Uganda to support the fight against HIV in rural areas and work towards ending the HIV epidemic by 2030.

## Supporting information

**S1 File. Data comprised of transcripts and codebook.**
(DOCX)

## Acknowledgments

The authors also acknowledge the parents, guardians, teachers, and head teachers of the participating secondary schools for their contribution to the data collected in this study. The authors

gratefully acknowledge Robens Mutatina (RM), Edison Byamugisha (EB) and Justine Kemigisha (JK) the Research Assistants who fully participated in the collection of the data.

## Author Contributions

**Conceptualization:** Paul Waswa Ssali, Timothy Mwanje Kintu, Immaculate Karungi, Agnes Kisakye Namuyaba, Tonny Kyagambiddwa, Ruth Namaseruka, Mark Agaba, Celestino Obua, Edith K. Wakida, Jerome Kahuma Kabakyenga.

**Data curation:** Paul Waswa Ssali, Timothy Mwanje Kintu, Immaculate Karungi, Agnes Kisakye Namuyaba, Tonny Kyagambiddwa, Ruth Namaseruka, Mark Agaba, Jerome Kahuma Kabakyenga.

**Formal analysis:** Paul Waswa Ssali, Timothy Mwanje Kintu, Immaculate Karungi, Agnes Kisakye Namuyaba, Tonny Kyagambiddwa, Ruth Namaseruka, Mark Agaba, Jerome Kahuma Kabakyenga.

**Funding acquisition:** Paul Waswa Ssali, Immaculate Karungi, Agnes Kisakye Namuyaba, Mark Agaba, Celestino Obua, Edith K. Wakida.

**Investigation:** Paul Waswa Ssali, Timothy Mwanje Kintu, Agnes Kisakye Namuyaba, Tonny Kyagambiddwa, Ruth Namaseruka, Mark Agaba, Jerome Kahuma Kabakyenga.

**Methodology:** Paul Waswa Ssali, Timothy Mwanje Kintu, Immaculate Karungi, Agnes Kisakye Namuyaba, Tonny Kyagambiddwa, Ruth Namaseruka, Mark Agaba, Jerome Kahuma Kabakyenga.

**Project administration:** Paul Waswa Ssali, Immaculate Karungi, Jerome Kahuma Kabakyenga.

**Resources:** Paul Waswa Ssali, Celestino Obua.

**Supervision:** Celestino Obua, Edith K. Wakida, Jerome Kahuma Kabakyenga.

**Validation:** Paul Waswa Ssali, Timothy Mwanje Kintu, Immaculate Karungi, Agnes Kisakye Namuyaba, Tonny Kyagambiddwa, Ruth Namaseruka, Mark Agaba, Celestino Obua, Edith K. Wakida, Jerome Kahuma Kabakyenga.

**Visualization:** Paul Waswa Ssali, Timothy Mwanje Kintu, Immaculate Karungi, Agnes Kisakye Namuyaba, Tonny Kyagambiddwa, Ruth Namaseruka, Mark Agaba, Celestino Obua, Edith K. Wakida, Jerome Kahuma Kabakyenga.

**Writing – original draft:** Paul Waswa Ssali, Timothy Mwanje Kintu, Immaculate Karungi, Agnes Kisakye Namuyaba, Tonny Kyagambiddwa, Ruth Namaseruka, Mark Agaba, Celestino Obua, Edith K. Wakida, Jerome Kahuma Kabakyenga.

**Writing – review & editing:** Paul Waswa Ssali, Timothy Mwanje Kintu, Immaculate Karungi, Agnes Kisakye Namuyaba, Tonny Kyagambiddwa, Ruth Namaseruka, Mark Agaba, Celestino Obua, Edith K. Wakida, Jerome Kahuma Kabakyenga.

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
