## [Decision Letter · Decision Letter 0]

22 Jan 2024

PGPH-D-23-02018

'If you find that I am HIV Positive, Don't Tell me': Exploring the barriers and recommendations for HIV Prevention services utilization among youth in rural southwestern Uganda

Dear Dr. Kabakyenga,

Thank you for submitting your manuscript to PLOS Global Public Health. After careful consideration, we feel that it has merit but does not fully meet PLOS Global Public Health’s publication criteria as it currently stands. Therefore, we invite you to submit a revised version of the manuscript that addresses the points raised during the review process.

Please note that we have only been able to secure a single reviewer to assess your manuscript. We are issuing a decision on your manuscript at this point to prevent further delays in the evaluation of your manuscript. Please be aware that the editor who handles your revised manuscript might find it necessary to invite additional reviewers to assess this work once the revised manuscript is submitted. However, we will aim to proceed on the basis of this single review if possible. 

The reviewer has suggested a number of revisions to improve the quality of reporting and contextualization of this study. Could you please carefully address all the concerns raised?

We look forward to receiving your revised manuscript.

Kind regards,

Marianne Clemence

Staff Editor

Journal Requirements:

Additional Editor Comments (if provided):

Reviewers' comments:

Reviewer's Responses to Questions

**Comments to the Author**

1. Does this manuscript meet PLOS Global Public Health’s publication criteria? Is the manuscript technically sound, and do the data support the conclusions? The manuscript must describe methodologically and ethically rigorous research with conclusions that are appropriately drawn based on the data presented.

Reviewer #1: Yes

2. Has the statistical analysis been performed appropriately and rigorously?

Reviewer #1: N/A

3. Have the authors made all data underlying the findings in their manuscript fully available (please refer to the Data Availability Statement at the start of the manuscript PDF file)?

Reviewer #1: Yes

4. Is the manuscript presented in an intelligible fashion and written in standard English?

Reviewer #1: Yes

5. Review Comments to the Author

Reviewer #1: This manuscript describes a qualitative assessment of barriers to HIV prevention services in the Rubirizi district of Uganda. The authors conducted focus groups and interviews with rural youth, teachers, and health care personnel, and they reported barriers at the individual, community, and health care level. They additionally solicited recommendations for improving HIV prevention services among youth, which included increased peer support, awareness campaigns, and economic empowerment initiatives. While the qualitative findings aren’t necessarily novel (many have already been reported in the existing literature about youth and HIV prevention in related contexts), the interviews are done with a focus on an important population in a rural context. With some revision, the manuscript could make a useful contribution to the HIV prevention literature in describing barriers to HIV prevention services.

Major considerations:

1. Were questions about PrEP included in these interviews? As one of the primary biomedical HIV prevention strategies, the absence of any content related to PrEP stands out as an important gap that was either not addressed in the interviews/focus groups or if asked, not included in the manuscript. This should be mentioned in the limitations section if it wasn’t included.

2. Results, recommendations section: Were youth participants asked about how to overcome barriers to accessing HIV prevention services? If so, did they corroborate any of the recommendations included? All of the quotations provided in the recommendations section were from key informants or in-depth interviews only (presumably all of whom are adults). The voices of the youth participants in focus groups would lend additional strength to the recommendations, since they are the primary focus of this study.

Minor considerations:

3. There are a number of grammatical or typographical errors throughout the manuscript (e.g., pg 4 line 97 “finishing villages” should read “fishing villages”). I didn’t highlight further examples here but would suggest a careful review of the manuscript to correct them.

4. Pg 4, lines 61-65: This sentence cites a reference (4) that details programming put in place in South Africa. Is there a different reference that specifically describes some of the HIV prevention strategies put in place in Uganda, which would be more relevant to the context of this study?

5. Pg 4, lines 83-84: As best I can tell, this study doesn’t attempt to differentiate between barriers for youth in school vs. out of school. This sentence can probably be removed if that is the case.

6. Pg 4, lines 86-89: The objective of this study could use further clarification. The statement only describes exploring barriers to utilization of HIV prevention services and recommendations but doesn’t mention that this study is focused on barriers to HIV prevention services for rural youth in Uganda specifically. The opening sentence of the discussion section is more specific as to the study objective.

7. Pg 5, lines 112-115: I’m not especially familiar with protections for research participants in Uganda. Is it accepted practice that a head teacher can provide informed consent on behalf of a participant <18 years old in place of a parent or legal guardian? This doesn’t need to be added to the manuscript specifically.

8. Pg 5, line 126: If youth participants and teachers were only selected from 4 schools in the district, it would appear that students and teachers from those 4 schools in the convenience sample were eligible, rather than all youth aged 15-24 in the district. Please clarify whether I’ve understood this correctly.

9. Table 1: could the authors clarify what the position “ART in-charge” means? While ART is an abbreviation I’m familiar with and that the authors have defined previously in the manuscript, the position name is still a bit confusing to me. Is this the person in charge of the ART program for a village?

10. Table 2: please provide column percentages for each cell.

11. Pg 11, lines 204-205: How the authors define barriers for the purpose of the study is probably more appropriate for the methods.

12. Discussion: In the introduction and methods, there was an emphasis on the fact that this study was done among rural youth in a district with many fishing villages (to emphasize the uniqueness of the context, I presume). However, the findings and discussion don’t really highlight whether the rural or fishing village context impacts the findings or whether any of the participants acknowledged that some of these barriers are in part because they live/work in a rural area (except for maybe the recommendation to expand awareness campaigns to the village level, pg 17 line 339-342). While not mandatory, contextualizing how some of the qualitative findings either compare to findings from urban settings or how delivering services in a rural setting might involve different barriers could be a useful addition where relevant.

6. PLOS authors have the option to publish the peer review history of their article (what does this mean?). If published, this will include your full peer review and any attached files.

**Do you want your identity to be public for this peer review?** For information about this choice, including consent withdrawal, please see our Privacy Policy.

Reviewer #1: **Yes: **Robert A. Bonacci

---

## [Decision Letter · Decision Letter 1]

24 Apr 2024

PGPH-D-23-02018R1

'If you find that I am HIV Positive, Don't Tell me': Exploring the barriers and recommendations for HIV Prevention services utilization among youth in rural southwestern Uganda

Dear Dr. Kabakyenga,

Thank you for submitting your manuscript to PLOS Global Public Health. After careful consideration, we feel that it has merit but does not fully meet PLOS Global Public Health’s publication criteria as it currently stands. Therefore, we invite you to submit a revised version of the manuscript that addresses the points raised during the review process.

EDITOR:

Requesting you to kindly address the following in your manuscript-

1. Please indicate the theoretical model used in your study

2. How did you decide on the number of FGDs and ID interviews to be conducted?

3. Please indicate the members of your team who were involved in the data collection and analysis in the methods section using their initials as well as what they brought to the study,

4. Please note that in the qualitative analysis the codes cannot be directly abstracted into subthemes- they need to abstracted, and categorised. the categories need to be mapped under subthemes and themes. Hence, you seem to have missed some steps in the analysis.

5. Kindly address trustworhiness in your research- i.e., credibility, confirmability, dependability and transferability in terms of qualitative research. Also, please add a note on reflexivity within your study.

6. Kindly work on categorising your codes and deriving subthemes and themes. The manuscript currently seems to have too many themes and would benefit from categorisation prior to arriving at these themes. Also, inference under each theme needs to be more comprehensive- the themes currently comprise quotes and very little inference.

7. All quotes in the manuscript have a suffix- 03/2022, it is unclear what this means. You could clarify in the methods how you have denoted the quotes.

8. A table with examples of coding, abstrcation, categorisation, arriving at subthemes and themes would be helpful.

We look forward to receiving your revised manuscript.

Kind regards,

Rashmi Josephine Rodrigues, M.D., Ph.D.

Academic Editor

Journal Requirements:

1. Please ensure that the Title in your manuscript file and the Title provided in your online submission form are the same.

Additional Editor Comments (if provided):

Reviewers' comments:

Reviewer's Responses to Questions

**Comments to the Author**

1. If the authors have adequately addressed your comments raised in a previous round of review and you feel that this manuscript is now acceptable for publication, you may indicate that here to bypass the “Comments to the Author” section, enter your conflict of interest statement in the “Confidential to Editor” section, and submit your "Accept" recommendation.

Reviewer #1: All comments have been addressed

2. Does this manuscript meet PLOS Global Public Health’s publication criteria? Is the manuscript technically sound, and do the data support the conclusions? The manuscript must describe methodologically and ethically rigorous research with conclusions that are appropriately drawn based on the data presented.

Reviewer #1: Yes

3. Has the statistical analysis been performed appropriately and rigorously?

Reviewer #1: Yes

4. Have the authors made all data underlying the findings in their manuscript fully available (please refer to the Data Availability Statement at the start of the manuscript PDF file)?

Reviewer #1: Yes

5. Is the manuscript presented in an intelligible fashion and written in standard English?

Reviewer #1: Yes

6. Review Comments to the Author

Reviewer #1: I thank the authors for their attentiveness to my prior feedback. I have only the following minor additional suggestions below.

1. Pg 11, line 237: I believe there may be a typo or incomplete sentence “…barriers to utilization of HIV prevention services emerged during discussions and teachers with youth and teachers in charge of health schools.”

2. Pg 18, line 405: sentence should begin with a capital letter.

7. PLOS authors have the option to publish the peer review history of their article (what does this mean?). If published, this will include your full peer review and any attached files.

**Do you want your identity to be public for this peer review?** For information about this choice, including consent withdrawal, please see our Privacy Policy.

Reviewer #1: No

---

## [Editor Report · Decision Letter 2]

12 Aug 2024

'If you find that I am HIV Positive, Don't Tell me': Exploring the barriers and recommendations for HIV Prevention services utilization among youth in rural southwestern Uganda

PGPH-D-23-02018R2

Dear Dr. Kabakyenga,

We are pleased to inform you that your manuscript ''If you find that I am HIV Positive, Don't Tell me': Exploring the barriers and recommendations for HIV Prevention services utilization among youth in rural southwestern Uganda' has been provisionally accepted for publication in PLOS Global Public Health.

Best regards,

Julia Robinson

Executive Editor